# DP-LFlow: Differentially Private Latent Flow for Scalable Sensitive Image Generation

**Dihong Jiang**                                                                 *dihong.jiang@uwaterloo.ca*
*Department of Computer Science*
*University of Waterloo*

**Sun Sun**                                                                      *sun.sun@nrc-cnrc.gc.ca*
*National Research Council Canada*
*University of Waterloo*

**Reviewed on OpenReview:** *https://openreview.net/forum?id=GEcneTl9Mk*

## Abstract

Privacy concerns grow with the success of modern deep learning models, especially when the training set contains sensitive data. Differentially private generative model (DPGM) can serve as a solution to circumvent such concerns by generating data that are distributionally similar to the original data yet with differential privacy (DP) guarantees. While GAN has attracted major attention, existing DPGMs based on flow generative models are limited and only developed on low-dimensional tabular datasets. The capability of *exact* density estimation makes the flow model exceptional when density estimation is of interest. In this work, we will first show that it is challenging (or even infeasible) to train a DP-flow via DP-SGD, i.e. the workhorse algorithm for private deep learning, on high-dimensional image sets with acceptable utility, and then we give an effective solution by reducing the generation from the pixel space to a lower dimensional latent space. We show the effectiveness and scalability of the proposed method via extensive experiments, where the proposed method achieves a significantly better privacy-utility trade-off compared to existing alternatives. Notably, our method is the first DPGM to scale to high-resolution image sets (up to $256 \times 256$). Our code is available at `https://github.com/dihjiang/DP-LFlow`.

## 1 Introduction

Large-scale datasets (Deng et al., 2009; Lewis et al., 2004; Bennett et al., 2007) facilitate the great success of modern machine learning (ML) systems. However, privacy concerns arise especially when sensitive data (e.g. human face images and medical data) are involved in the training. Prior privacy-preserving techniques include naive data anonymization (Narayanan & Shmatikov, 2008), $k$-anonymity (Sweeney, 2002), $l$-diversity (Machanavajjhala et al., 2007), semantic security (Goldwasser & Micali, 1984), and differential privacy (DP) (Dwork, 2006), where DP is recognized as a rigorous quantization of privacy, and it has become the gold-standard privacy-preserving technique in the current ML community.

Differentially private generative model (DPGM) aims to synthesize data that are distributionally similar to the private data while satisfying DP guarantees so that the individual privacy that can be inferred from the incremental change in the dataset is bounded. DPGM can (1) serve as a proxy for releasing private data, and can (2) generate data for private data analysis tasks (e.g. data querying and ML tasks) without incurring further privacy cost, as ensured by the post-processing theorem (Dwork et al., 2014).

Generative adversarial network (GAN) (Goodfellow et al., 2014) has attracted the most attention in developing DPGMs (Xie et al., 2018; Torkzadehmahani et al., 2019; Jordon et al., 2019; Long et al., 2021; Augenstein et al., 2020; Chen et al., 2020), and VAE also draws a lot of attention (Chen et al., 2018; Acs et al., 2018; Takahashi et al., 2020; Pfitzner & Arnrich, 2022; Weggenmann et al., 2022; Bernau et al., 2022).

In contrast, the DPGMs based on the normalizing flow are relatively limited (Waites & Cummings, 2021; Lee et al., 2022). Due to the capability of the *exact* density computation, flow models are particularly useful when density is of interest in some applications. For example, developing a DP-flow could benefit performing anomaly detection in a privacy-preserving manner (Waites & Cummings, 2021). However, both existing DP-flow works (Waites & Cummings, 2021; Lee et al., 2022) are restricted to tabular datasets (with lower data dimensions). It remains unclear whether DP-flow is feasible on higher-dimensional datasets such as image datasets.

Abadi et al. (2016) propose Differentially Private Stochastic Gradient Descent (DP-SGD) algorithm, which has become the standard approach to train a DP deep learner. The core steps of DP-SGD are clipping the per-example gradient norm and injecting Calibrated Gaussian noise into the aggregated gradient of a batch. Briefly, let $p$ denote the model dimension, the noise introduced by DP-SGD in each iteration is given by $\mathbf{z} \sim \mathcal{N}(\mathbf{0}_p, \sigma^2 C^2 \mathbb{I}_{p \times p})$, where $C$ is the clipping bound of gradient norm and $\sigma$ is a noise multiplier. Apparently, $\mathbb{E}[\|\mathbf{z}\|_2^2] = p\sigma^2 C^2 \propto p$, which means the model utility may not be preserved with DP-SGD for a large model. For example, Yu et al. (2021) show that the gradient will be submerged in the added noise (by DP-SGD) when the model becomes larger on a series of ResNet (He et al., 2016) variants. This utility drop could become more pronounced under strong privacy guarantees (e.g. $\epsilon = 1$), as the noise multiplier becomes larger. Note that according to $\epsilon$-DP definition (given in Section 2), the privacy protection becomes weak when $\epsilon = 10$ or above, because the ratio of two probabilities is upper bounded by $\exp(10) \approx 2.2 \times 10^4$, whereas they are presumed to be comparable for practical deployment (e.g. $\epsilon = 1$ or below). However, a line of recent SoTA DPGMs only generate acceptable images at $\epsilon = 10$, whereas they are not able to scale well on small $\epsilon$ (e.g. $\epsilon = 1$) (Chen et al., 2020; Jordon et al., 2019; Torkzadehmahani et al., 2019).

Training a normalizing flow with DP-SGD on an image set seems ostensibly easy, but any ML researcher/engineer will encounter the two non-trivial empirical challenges, as also pointed out by Lee et al. (2022): (1) *batch normalization (BN) challenge*: flow models usually apply a batch normalization (BN) layer in each block to boost the performance, where the per-example gradient (in DP-SGD) is not available, as BN will break the independence among all gradients in a batch; (2) *model complexity challenge*: flow models generally consist of repetitive blocks of invertible transformations with a large depth, which results in higher model complexity compared to other generative models. We explore the challenges with two SoTA flow models, i.e. RealNVP (Dinh et al., 2017) and Glow (Kingma & Dhariwal, 2018). Glow uses activation normalization as an alternative to BN, thus per-example gradient is computable (i.e. no BN challenge for Glow). However, both RealNVP and Glow suffer from the model complexity challenge, i.e. they tend to be more complex than other generative model counterparts for synthesizing images with similar fidelity, thus are more challenging to attain desirable model utility with DP-SGD. Our motivating example is shown in Figure 6, where naively training a flow generative model in the pixel space with DP-SGD results in null model utility.

Our contributions can be summarized as follows:

- We explore the practical challenges of training a DP flow via DP-SGD on image sets, and propose an efficient and effective solution, i.e. differentially private latent flow (DP-LFlow), by reducing the training of flow from the full pixel space to a lower-dimensional latent space, which is more resilient to the noise perturbation (by DP-SGD).

- Training DPGM on high-resolution images ($256 \times 256$ pixels and beyond) is extraordinarily challenging due to the inevitably increased model dimension, and to our best knowledge, none of the existing related works attempt to do so. In this work, we will show that DP-LFlow is also scalable to the high-resolution image ($256 \times 256$) generation with DP constraints.

- The proposed method yields state-of-the-art (SoTA) performance on model utility under the same $(\epsilon, \delta)$-DP constraint on widely compared image benchmarks. Moreover, our method indicates more robustness and scalability on different datasets (gray-scale and RGB) and different DP constraints.

---

**Algorithm 1:** Gradient perturbation in DP-SGD

---

**Input:** Private training set $X = \{\mathbf{x}_i\}_{i=1}^N$, loss function $\mathcal{L}(\cdot)$, batch size $B$, noise multiplier $\sigma$, gradient clipping bound $C$, model parameter $\theta$

**1 for** $i \leftarrow 1$ ***to*** $B$ **do**

**2** $\quad$ $g_\theta(\mathbf{x}_i) = \nabla_\theta \mathcal{L}(\mathbf{x}_i; \theta)$

**3** $\quad$ $g_\theta(\mathbf{x}_i) = g_\theta(\mathbf{x}_i) \cdot \min\left(1, \frac{C}{\|g_\theta(\mathbf{x}_i)\|_2}\right)$

**4** $\tilde{g}_\theta = \frac{1}{B}\left[\sum_{i=1}^B g_\theta(\mathbf{x}_i) + \mathcal{N}(0, \sigma^2 C^2 \mathbb{I})\right]$

---

## 2 Preliminary

In this section, we recall background knowledge in differential privacy.

### 2.1 Differential Privacy

Differential privacy is widely regarded as a rigorous quantization of privacy, which upper-bounds the deviation in the output distribution of a randomized algorithm given an incremental deviation in the input. Formally, we have the following definition:

**Definition 2.1** (($\epsilon, \delta$)-DP (Dwork et al., 2014))**.** A randomized mechanism $\mathcal{M} : \mathcal{D} \to \mathcal{R}$ with domain $\mathcal{D}$ and range $\mathcal{R}$ satisfies ($\epsilon, \delta$)-differential privacy if for any two adjacent inputs $D, D' \in \mathcal{D}$ and for any subset of outputs $\mathcal{S} \subseteq \mathcal{R}$ it holds that

$$\Pr[\mathcal{M}(D) \in \mathcal{S}] \leq \exp(\epsilon) \cdot \Pr[\mathcal{M}(D') \in \mathcal{S}] + \delta \tag{1}$$

where adjacent inputs (a.k.a. neighbouring datasets) only differ in one entry. Particularly, when $\delta = 0$, we say that $\mathcal{M}$ is $\epsilon$-DP.

There is a convenient parallel composition theorem for $\epsilon$-DP mechanisms:

**Theorem 2.1** (Parallel composition theorem of $\epsilon$-DP, (McSherry, 2009))**.** Let $\mathcal{M}_i$ ($i = 1, 2, \ldots, k$) be $k$ DP mechanisms, and each $\mathcal{M}_i$ satisfies $\epsilon_i$-DP. Given a deterministic partitioning function $f$, let $D_1, D_2, \ldots, D_k$ be the disjoint partitions by executing $f$ on $D$. Releasing $\mathcal{M}_1(D_1), \ldots, \mathcal{M}_k(D_k)$ satisfies $\max_{i \in \{1,2,\ldots,k\}} \epsilon_i$-DP.

We will extend the above parallel composition to the ($\epsilon, \delta$)-DP notion in Section 3.

A famous theorem, i.e. post-processing theorem, which is utilized by existing works (as well as ours) for proving DP guarantee of a published model, is given by:

**Theorem 2.2** (Post-processing theorem, (Dwork et al., 2014))**.** If $\mathcal{M}$ satisfies ($\epsilon, \delta$)-DP, $F \circ \mathcal{M}$ will satisfy ($\epsilon, \delta$)-DP for any function $F$ with $\circ$ denoting the composition operator.

Sampling from a DPGM is independent of training data, thus can be viewed as a post-processing step and does not breach the DP guarantee.

Rényi differential privacy (RDP) extends ordinary DP using Rényi's $\alpha$ divergence (Rényi, 1961) and provides tighter and easier composition property than the ordinary DP notion, thus we adopt RDP to accumulate the privacy cost. We defer details of RDP and implementation to Appendix E.

### 2.2 DP-SGD

Within predetermined training iterations, in each iteration, DP-SGD (Abadi et al., 2016) subsamples a batch from the training set, clips and perturbs the gradient as in Algorithm 1, and optimizes the model with privatized gradient $\tilde{g}_\theta$. As mentioned earlier, the norm of the Gaussian noise introduced at line 4 in Algorithm 1 will scale linearly with the model dimension, thus will generally degrade the utility of large models.

### 2.3 Flow-based Generative Models

We briefly recap the flow generative models. Flow models learn a bijective map $\mathsf{T}$ between a simple prior distribution $q_0$ (e.g. Gaussian) and the target distribution $q$: $\mathbf{z} \sim q_0 \Leftrightarrow \mathsf{T}(\mathbf{z}) \sim q$. Through the change-of-variable formula, the log-likelihood of input is tractable:

$$q_\mathsf{T}(\mathbf{x}) = q_0(\mathbf{z}) \left| \det \frac{d\mathbf{z}}{d\mathbf{x}} \right| = q_0\big(\mathsf{T}^{-1}(\mathbf{x})\big) \left| \det \frac{d\mathsf{T}^{-1}(\mathbf{x})}{d\mathbf{x}} \right| \tag{2}$$

Parameterize the bijective map by neural networks, we can train flow models by minimizing the Kullback–Leibler (KL) divergence between the true and estimated distribution:

$$\min_\mathsf{T} \mathbb{D}_{KL}(q(\mathbf{x})\|q_\mathsf{T}(\mathbf{x})) = \min_\mathsf{T} \int q(\mathbf{x}) \log \frac{q(\mathbf{x})}{q_\mathsf{T}(\mathbf{x})} d\mathbf{x} \tag{3}$$

$$= \min_\mathsf{T} \mathbb{E}_{\mathbf{x} \sim q(\mathbf{x})}[-\log q_\mathsf{T}(\mathbf{x})] - \mathbb{H}[q] \tag{4}$$

where $\mathbb{H}[q]$ is the entropy of true distribution. Therefore, training the flow model amounts to minimizing the negative log-likelihood of input.

## 3 Method: DP-LFlow

As shown in (Yu et al., 2021), a smaller/simpler ResNet is more resilient to DP-SGD. The intuition is that the model utility will saturate as the model complexity increases. Therefore, an over-complicated model may (unnecessarily) be disturbed by more noises in DP-SGD. We first confirm this insight for DPGMs as well. Consider privately training a VAE on MNIST (LeCun et al., 1998) for example. Figure 1 shows such comparison under non-private and private settings (with different privacy costs), respectively, where the generation quality is measured by Fréchet Inception Distance (FID) (Heusel et al., 2017) (lower is better). Figure 1 indicates that a smaller VAE generates better images than larger counterparts (both qualitatively and quantitatively) under the DP training, even though larger VAEs perform better in the non-DP setting. In fact, shrinking the model size under the DP training will benefit from the following aspects: (1) smaller models are more resilient to (larger) noises (associated with strong DP guarantees); (2) a significant training time overhead remains a notable challenge for DP-SGD (Subramani et al., 2021) due to the gradient clipping and randomization. Smaller models could facilitate more efficient DP training; (3) as suggested by prior work (De et al., 2022), a larger batch size in DP-SGD contributes to better model utility. With limited GPU memory, smaller models allow for a large training batch size, since both input and model are sent to GPU for training in practice.

**Latent Flow:** However, the model expressiveness will be restricted if the model is too simple. Therefore, we aim to design a model that is small yet expressive enough so that we can achieve a better privacy-utility trade-off with DP-SGD. Inspired by the recent latent diffusion model (Rombach et al., 2022) that achieves SoTA text-to-image generation performance via reducing the diffusion process from the raw input space to a lower dimensional latent space, we propose to train a normalizing flow in a similar manner (i.e. on the latent code produced by an autoencoder), by simultaneously minimizing the reconstruction loss of the autoencoder and the negative log-likelihood of the flow. Specifically, let $\phi, \theta, \omega$ parameterize the encoder, decoder, and flow, respectively, and we adopt the notations in Section 2.3. Let $\mathbf{w} = f_\phi(\mathbf{x})$ be the latent embedding of the autoencoder, the training objective can be written as:

$$\min_{\phi,\theta,\omega} \mathbb{E}_{\mathbf{x} \sim q}[\underbrace{\|\mathbf{x} - D_\theta(\mathbf{w})\|_2^2}_{\text{Autoencoder loss}} \times T^2 \underbrace{-\big(\log q_0(\mathbf{z}) + \log \left| \det \frac{d\mathsf{T}_\omega^{-1}(\mathbf{w})}{d\mathbf{w}} \right|\big)}_{\text{Normalizing flow loss}}] \tag{5}$$

where $\mathbf{z} \sim q_0 = \mathcal{N}(0, \mathbb{I})$ and $T$ is a temperature parameter. As shown in Rombach et al. (2022), the semantic meaning of most images still remains after aggressive compression, thus allowing us to train a flow in an aggressively trimmed latent space, which avoids unnecessary and expensive computation on full input dimensions. It is worth mentioning that latent flow is also not sensitive to BN layers (e.g. for RealNVP), i.e.

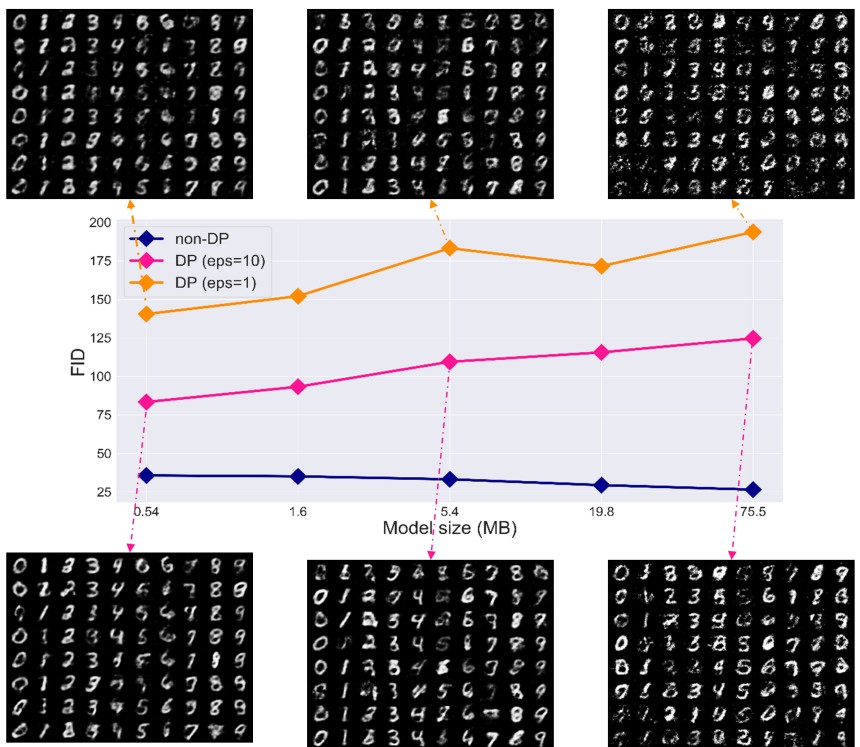

Figure 1: FID vs. VAE size (in MB). We only vary the model complexity, with all the rest training parameters (e.g. subsampling rate, noise multiplier, training iterations) fixed.

the utility of latent flow is slightly reduced by removing the BN layer, which validates the use of DP-SGD for latent flow models. Under the non-private setting, the proposed model with a configuration of size 2.5 MB achieves an FID = 12.5 on MNIST, which is already superior to VAE counterparts with a larger model size (e.g. in Figure 1), suggesting its promising usage under the private training.

**Partitioning Dataset:** Current SOTA methods tend to apply conditional generative models (conditioned on labels), where the label is encoded in the model as part of the input, thus the noise perturbation also distorts the label information, which is unnecessary. We run DP-SGD on the proposed model under the conditional setting (autoencoder + conditional flow), and observe that the label information is largely distorted when $\epsilon = 1$ (see Section 4.5 for details). To circumvent the perturbation to labels, we propose to *partition the dataset according to labels*, train unconditional generative models on each of the subsets, and release the union of all unconditional generators as the resulting model. The partitioning is also beneficial for shrinking the model size, as each generator is only interactive with a sole data modality instead of multi-modalities. Adapted from proposition 2.5 in (Li et al., 2016), the DP guarantee for the union can be derived from the parallel composition, by extending the original parallel composition theorem (Theorem 2.1) from $\epsilon$-DP notion to $(\epsilon, \delta)$-DP notion. The proof can be found in Appendix A.

**Theorem 3.1.** Let $\mathcal{M}_i$ $(i = 1, 2, \ldots, k)$ be $k$ DP mechanisms, and each $\mathcal{M}_i$ satisfies $(\epsilon_i, \delta_i)$-DP. Given a deterministic partitioning function $f$, let $D_1, D_2, \ldots, D_k$ be the disjoint partitions by executing $f$ on $D$. Releasing $\mathcal{M}_1(D_1), \ldots, \mathcal{M}_k(D_k)$ satisfies $(\max_{i \in \{1,2,\ldots,k\}} \epsilon_i, \max_{i \in \{1,2,\ldots,k\}} \delta_i)$-DP.

For simplicity, we set $\epsilon_i, \delta_i$ the same as the target $\epsilon, \delta$ for all sub-models in the experiment. The schematic workflow of DP-LFlow is shown in Figure 7.

## 4 Experiments

In this section, we evaluate and compare DP-LFlow against SoTA baselines through extensive experiments in Section 4.2. More importantly, we will show that DP-LFlow is amenable to high-resolution image sets in Section 4.3, which was hardly studied in prior related works. Besides, since flow models are capable of computing the exact likelihood of input, they are sometimes more applicable when density estimation is of concern. Section 4.4 shows such an example, where DP-LFlow can detect intra-dataset anomaly with DP guarantees. RealNVP is used as the flow model in DP-LFlow, as it yields better performance in practice. Implementation details and neural network configurations are given in Appendix E.

### 4.1 Experimental Setup

**Datasets:** We consider three widely used image datasets, including both grayscale images (MNIST (LeCun et al., 1998), Fashion MNIST (Xiao et al., 2017)) and RGB images (CelebA (Liu et al., 2015)), as well as one high-resolution RGB datasets (CelebA-HQ (Karras et al., 2018), for our presentation only). For MNIST and Fashion MNIST, we generate images conditioned on 10 respective labels. For CelebA and CelebA-HQ, we condition on gender. Descriptions and preprocessing of the datasets are given in the Appendix B.

**Evaluation Tasks & Metrics:** We evaluate and compare DPGMs by two metrics via 60k generated images:

- Fréchet Inception Distance (FID) (Heusel et al., 2017).

- Classification accuracy. We train three different classifiers, e.g. logistic regression (LR), multi-layer perceptron (MLP), and convolutional neural network (CNN), on generated images, then test the classifier on real images. We take 5 runs and report the average. See Appendix E for details of classifiers.

**Post-processing:** We observe that FID is quite sensitive to the noise around the object (e.g. digits, cloths) and the sharpness of images, even when the semantic content is retained. We apply a post-processing step to our DP image generation by first smoothing and then sharpening the generated images, which could lead to ~10 improvement in FID. Whereas this post-processing does not improve classification accuracy too much. Adjusting the blurriness and sharpness of generation is not involved with training data, thus is considered a post-processing step that does not breach the DP guarantee by Theorem 2.2. We will report our quantitative evaluations with and without this trick.

**SoTA Baselines:** Our method is compared with the following baseline methods that are also developed on image datasets, i.e. DP-CGAN (Torkzadehmahani et al., 2019), DP-MERF (Harder et al., 2021), Datalens (Wang et al., 2021), PATE-GAN (Jordon et al., 2019), G-PATE (Long et al., 2021), GS-WGAN (Chen et al., 2020), DP-Sinkhorn (Cao et al., 2021). For more details, we refer readers to Section 6 and respective references.

### 4.2 Comparison with SoTA Baselines

We follow the general benchmark for evaluating DP generation, by considering both weak (e.g. $\epsilon = 10$) and strong (e.g. $\epsilon = 1$) privacy guarantees. The proposed DP-LFlow is compared with SoTA baselines through extensive qualitative and quantitative experiments on both grayscale and RGB image datasets. For completeness, we also report our quantitative results without the post-processing trick, where the relative rank of our method without the post-processing trick remains the same as with the trick.

### 4.2.1 Under the Weak DP Guarantee $\epsilon = 10$

Most existing works perform reasonably when $\epsilon = 10$, as shown in Figure 2. Nevertheless, as shown in Table 1, DP-LFlow achieves significant improvement in both FID and classification accuracy.

Table 1: Quantitative comparison on MNIST and Fashion MNIST given $(10, 10^{-5})$-DP. Acc denotes classification accuracy, which is shown in %. ↑ and ↓ refer to higher is better or lower is better, respectively. We use boldface for the best performance. Results of DP-CGAN, GS-WGAN, DP-Sinkhorn are cited from Cao et al. (2021). Results of G-PATE and DataLens are cited from their papers, respectively. Ours* denotes our result without the post-processing trick.

| Method | MNIST | | | | Fashion MNIST | | | |
|---|---|---|---|---|---|---|---|---|
| | FID ↓ | LR Acc ↑ | MLP Acc ↑ | CNN Acc ↑ | FID ↓ | LR Acc ↑ | MLP Acc ↑ | CNN Acc ↑ |
| Real data (non-private) | 1.6 | 92.2 | 97.5 | 99.3 | 2.5 | 84.5 | 88.2 | 90.8 |
| DP-CGAN | 179.2 | 60 | 60 | 63 | 243.8 | 51 | 50 | 46 |
| DP-MERF | 121.4 | 79.1 | 81.1 | 82.0 | 110.4 | 72.3 | 70.8 | 73.2 |
| G-PATE | 150.6 | - | - | 80.9 | 171.9 | - | - | 69.3 |
| DataLens | 173.5 | - | - | 80.7 | 167.7 | - | - | 70.6 |
| GS-WGAN | 61.3 | 79 | 79 | 80 | 131.3 | 68 | 65 | 65 |
| DP-Sinkhorn | 55.6 | 79.1 | 79.2 | 79.1 | 129.4 | 70.2 | 70.2 | 68.9 |
| Ours | **16.8** | **85.8** | **93.0** | **95.3** | **72.3** | **78.6** | **78.8** | **81.0** |
| Ours* | 25.4 | 85.1 | 92.4 | 94.8 | 80.8 | 78.1 | 78.4 | 80.7 |

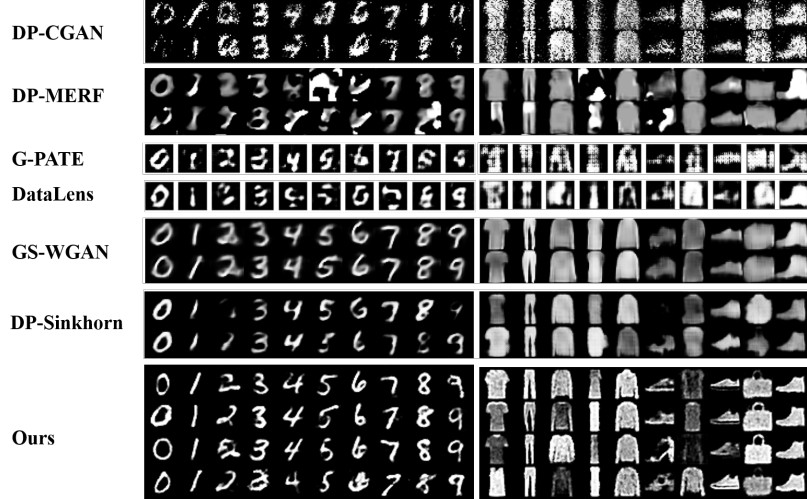

Figure 2: Qualitative comparison on MNIST and Fashion MNIST under $(10, 10^{-5})$-DP. Images of DP-CGAN, GS-WGAN, DP-Sinkhorn are cited from (Cao et al., 2021). Images of G-PATE and DataLens are cited from their papers, respectively.

### 4.2.2 Under the Strong DP Guarantee $\epsilon = 1$

When $\epsilon = 1$, i.e. the more challenging case under the DP setting, we consider both gray-scale image datasets and RGB image datasets for comparison. Figure 3 and Figure 4 show that the existing works are not readily amenable to a small $\epsilon$ such as 1. In contrast, DP-LFlow exhibits significant visual improvement on all three datasets.

Quantitatively, Table 2 indicates that DP-LFlow outperforms other baselines on CNN classification accuracy on all three datasets. Besides, DP-LFlow achieves FID scores that are comparable to SoTA performance on Fashion MNIST and achieves the best FID on MNIST and CelebA. Given that the three datasets are either largely different in input dimension or largely different in semantic content, the superior performance of DP-LFlow further validates its generality and versatility.

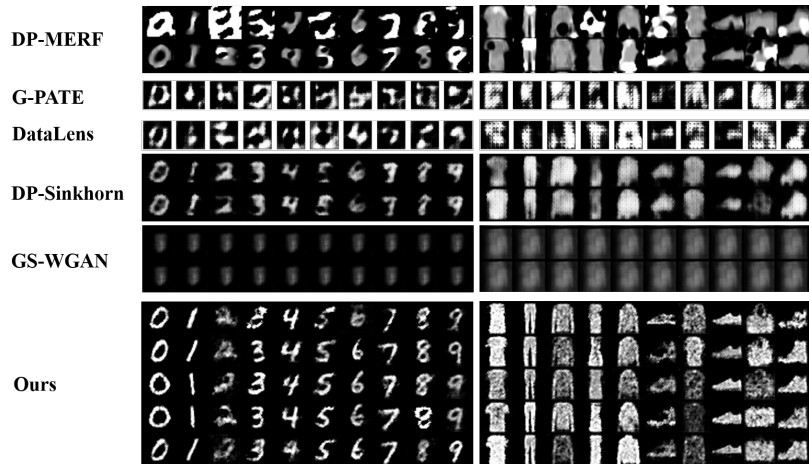

Figure 3: Qualitative comparison on MNIST and Fashion MNIST under $(1, 10^{-5})$-DP. Images of G-PATE and DataLens are cited from their papers, respectively.

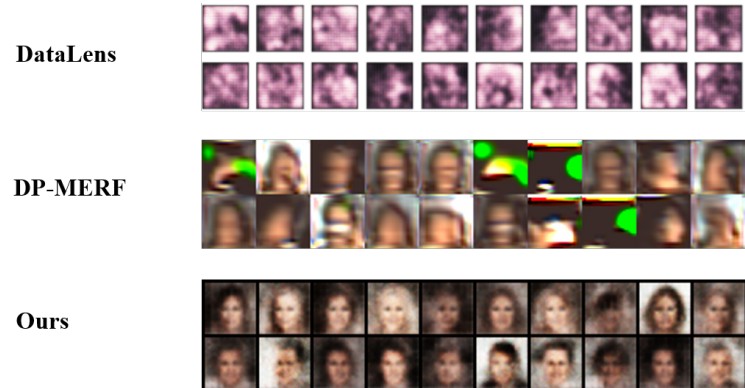

Figure 4: Qualitative comparison on CelebA conditioned on gender under $(1, 10^{-5})$-DP. Images of DataLens are cited from the original paper. Top row: female. Bottom row: male.

It is worth mentioning that a couple of recent works on DP diffusion models are also capable of generating realistic images under DP constraints and indicate superiority on downstream ML tasks (Dockhorn et al., 2023; Ghalebikesabi et al., 2023). For example, when $\epsilon = 1$, Dockhorn et al. (2023) achieve 93.4 and 73.6 test classification accuracies on MNIST and FMNIST, respectively, while ours are 88.9 and 77.4. However, their required computational resource is significantly higher than DP-LFlow, e.g. Dockhorn et al. (2023) need 8 GPUs and one day to train a DP diffusion model on MNIST and FMNIST, while our method only requires 1 single GPU and a few hours; On CelebA, Dockhorn et al. (2023) needs 8 GPUs and 4 days, while our method only needs 1 GPU and around half-day. More importantly, DP-LFlow still has two unique properties compared to DP diffusion models: (1) DP-LFlow can generate large images up to $3 \times 256 \times 256$ (see Section 4.3), while Dockhorn et al. (2023); Ghalebikesabi et al. (2023) only present generation results on images up to $3 \times 32 \times 32$; (2) DP-LFlow is able to conduct the DP-OOD detection task (see Section 4.4).

## 4.3 Generating High-resolution Images with DP Constraints

To our best knowledge, the highest resolution image dataset used in related works is CelebA downsampled at $64 \times 64$ pixels (by DataLens and G-PATE), which, however, is still not typically considered a high-resolution space. Here we consider a high-resolution dataset CelebA-HQ in $256 \times 256$, which is also close to many modern large-scale image classification benchmark datasets such as ImageNet ($224 \times 224$ or $256 \times 256$).

Table 2: Quantitative comparison on image datasets given $(1, 10^{-5})$-DP. FMNIST denotes Fashion MNIST. Acc denotes CNN classification accuracy, which is shown in %. ↑ and ↓ refer to higher is better or lower is better, respectively. We use boldface for the best performance. Results of PATE-GAN and G-PATE are cited from Long et al. (2021). Results of DataLens are cited from Wang et al. (2021). Ours* denotes our result without the post-processing trick.

| Dataset | Metrics | PATE-GAN | DP-MERF | DP-Sinkhorn | G-PATE | DataLens | Ours | Ours* |
|---------|---------|----------|---------|-------------|--------|----------|------|-------|
| MNIST | FID ↓ | 231.5 | 118.3 | 168.5 | 154.3 | 186.1 | **69.7** | 83.4 |
|  | Acc ↑ | 41.7 | 80.5 | 60.5 | 58.8 | 71.2 | **88.9** | 88.2 |
| FMNIST | FID ↓ | 253.2 | **104.2** | 184.2 | 214.8 | 195.0 | 132.4 | 143.5 |
|  | Acc ↑ | 42.22 | 73.1 | 62.1 | 58.12 | 64.8 | **77.4** | 76.8 |
| CelebA | FID ↓ | 434.5 | 219.4 | - | 293.2 | 297.7 | **204.8** | 217.7 |
|  | Acc ↑ | 44.48 | 57.6 | - | 70.2 | 70.6 | **73.2** | 72.1 |

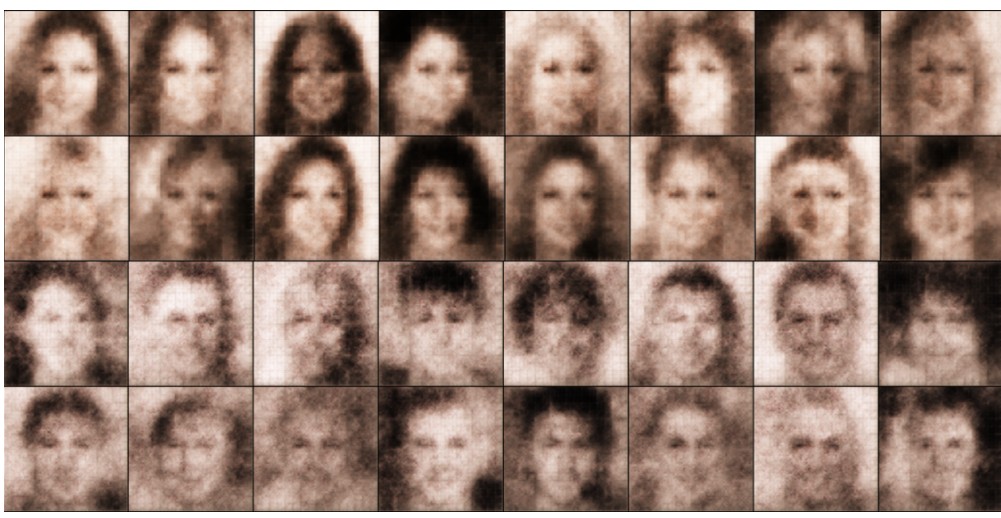

Figure 5: Samples from DP-LFlow trained on CelebA-HQ under $(10, 10^{-5})$-DP. Top 2 rows: female. Bottom 2 rows: male. FID = 328.6, LR classification accuracy = 77.4.

Note that now the input dimension drastically increases from $1 \times 28 \times 28 = 784$ (for MNIST and Fashion MNIST) to $3 \times 256 \times 256 \approx 1.97 \times 10^5$. This means that the model generally has to drastically scale up to adequately learn the input distribution, leading to a significant challenge for training a DPGM with DP-SGD.

Nevertheless, with the help of lower dimensional latent space where we can apply aggressive compression while retaining the semantic content, we can restrict the generative model to a size ($\sim 22$ MB, compared to $\sim 2.5$ MB on MNIST) that is suitable for DP training yet adequate to learn the data distribution. By training a DP-LFlow on CelebA-HQ, Figure 5 shows that DP-LFlow is able to produce diverse and recognizable face images with DP constraints on such high dimensional input space.

## 4.4 Differentially Private Out-of-distribution (OOD) Detection

One exceptional feature of flow models is their capability of *exact* density estimation, which can benefit certain applications such as out-of-distribution detection. A natural method to do so is to thresholding the log-likelihood, since it is presumed that the in-distribution (InD) data (where the training data are drawn from) will be assigned a higher likelihood than OOD data by the flow model. This idea can be readily extended to latent flow models by checking the log-likelihood of latent code mapped from the input data. As our sub-models are privately trained on each subset by class, we can immediately conduct the differentially

Table 3: Differentially private OOD detection results. The evaluation metric is shown by AUROC (higher is better, and 1.0 means InD and OOD likelihood are perfectly separable).

| InD class | MNIST | | Fashion MNIST | |
|---|---|---|---|---|
| | $\epsilon = 10$ | $\epsilon = 1$ | $\epsilon = 10$ | $\epsilon = 1$ |
| 0 | 0.98 | 0.87 | 0.90 | 0.87 |
| 1 | 0.99 | 0.98 | 0.97 | 0.96 |
| 2 | 0.86 | 0.75 | 0.90 | 0.88 |
| 3 | 0.88 | 0.82 | 0.91 | 0.91 |
| 4 | 0.95 | 0.85 | 0.87 | 0.85 |
| 5 | 0.90 | 0.75 | 0.91 | 0.86 |
| 6 | 0.97 | 0.77 | 0.84 | 0.82 |
| 7 | 0.93 | 0.88 | 0.97 | 0.97 |
| 8 | 0.89 | 0.74 | 0.86 | 0.79 |
| 9 | 0.95 | 0.83 | 0.94 | 0.93 |
| Average | 0.93 | 0.82 | 0.91 | 0.88 |

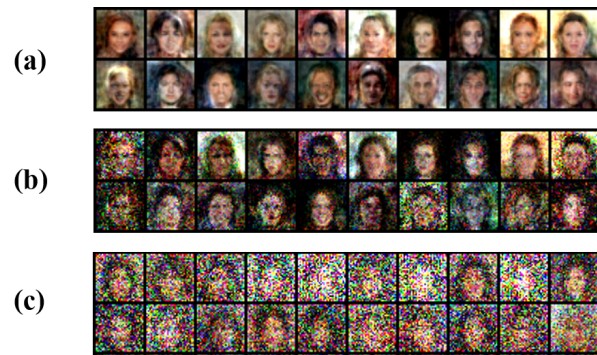

Figure 6: Training a RealNVP ($\sim$150 MegaBytes (MB)) on CelebA (a) non-privately with BN layers, (b) non-privately without BN layers, (c) under $(10, 10^{-5})$-DP without BN layers

private intra-dataset OOD detection, by treating each training class as the InD and the rest classes combined as OOD. A lower log-likelihood of the latent code computed from input data implies OOD-ness. We use Area Under Receiver Operating Curve (AUROC) as the evaluation metric, which is threshold-free and widely used in OOD detection literature. Table 3 indicates that under weak and strong guarantees, DP-LFlow can effectively detect intra-dataset OOD input on both MNIST and Fashion MNIST across all InD classes in a privacy-preserving manner.

### 4.5 Ablation Study

DP-LFlow contains two main components, and we conduct an ablation study as follows.

#### 4.5.1 Flow in the Pixel Space vs. Latent Space

We first trained a RealNVP with DP-SGD by removing the BN layer. Figure 6 show that the generation from a DP-RealNVP (subfigure (c)) is largely submerged in the noise. Besides, we also trained a Glow with DP-SGD (with various model complexities), yet ended up with null synthesis when targeting $(10, 10^{-5})$-DP. Both RealNVP and Glow suffer from the model complexity challenge. Compared to the generation of DP-LFlow in Figure 4 (which even targets a stronger privacy regime, i.e. $\epsilon = 1$), training a flow in the latent space relieves the model complexity challenge, thus is more resilient to the noise perturbation introduced by DP-SGD.

#### 4.5.2 With vs. Without Partitioning the Dataset

Qualitatively, we observe that training a conditional DP-LFlow on the whole dataset will fail to correctly condition on each label, (especially under $\epsilon = 1$), i.e. the generations conditioned on different labels are not consistent with the true labels, thus the numerical classification accuracy is relatively lower, although FID is fairly comparable. We summarize the numerical comparison in Table 4.

The reason is that, in contrast to uninvertible models like conditional GAN and conditional VAE that only encode labels in the latent vector, the label information in the conditional flow is encoded in all coupling layers, because flow is an invertible model so that both forward and backward pass needs label information. Thus, training conditional flow with DP-SGD will distort the label information more than uninvertible models. We also tried to modify the way of encoding labels in different variants: only encoding labels in the first, last, or both first and last coupling layers, or only encoding labels in the latent vector, but none

Table 4: Ablation on partitioning the MNIST dataset. We compare the performance without post-processing tricks (smoothing and sharpening) for simplicity.

| | $\epsilon$ | FID ↓ | LR Acc ↑ | MLP Acc ↑ | CNN Acc ↑ |
|---|---|---|---|---|---|
| With partitioning | 10 | 25.4 | 85.2 | **92.7** | **95.0** |
| Without partitioning | 10 | **21.3** | **86.2** | 91.5 | 91.8 |
| With partitioning | 1 | **83.4** | **83.2** | **82.4** | **88.3** |
| Without partitioning | 1 | 87.1 | 40.8 | 72.5 | 77.5 |

of the variants solve the issue. Instead, resorting to partitioning the dataset can readily bypass the label distortion in the private training, which leads to better model utility compared to a single conditional model on all classes, and by Theorem 3.1 we can retain the same level of privacy guarantees as publishing a single conditional generator. In other words, partitioning the dataset contributes to improving the privacy-utility trade-off.

## 5 Discussion

**Pretraining the autoencoder:** Though our current training proceeds by simultaneously updating the autoencoder and flow, one may wonder whether we can pretrain an autoencoder first and then train a flow with the pretrained autoencoder. However, there are a few concerns that impede the two-stage training paradigm:

- **Pretraining on a public dataset:** The majority of existing works (including all of our compared works) do not use external public dataset for pretraining, so it would make the comparison unfair if we do so. Besides, we need to put a strong assumption that the public dataset does not have any privacy concerns. While this can be easily ensured in experiments, it may not always be true in reality, thus is likely to undermine the potential impact of this work. Therefore, we stick to restricting our information to the private dataset only.

- **Pretraining on the private dataset:** In this case, changing a point in the dataset will result in a different pretrained autoencoder, which means all the gradients in a batch (when running DP-SGD on flow) will change, instead of at most one gradient is affected in original DP-SGD, thus the sensitivity of the gradient will scale up by $B$, i.e. the batch size. To make the same DP guarantee hold, we can recalculate the $\epsilon$ by replacing the original noise multiplier $\sigma$ with $\sigma \cdot B$ to account for the gradient sensitivity change in a minibatch. However, the actual noise multiplier is much larger and will undoubtedly destroy the model utility.

**Batch normalization (BN) vs. Group normalization (GN):** Replacing BN with GN is a common practice for training DP models, e.g. as in Luo et al. (2021). However, unlike the image classification network (Luo et al., 2021) that only takes forward pass, flow generative models are invertible models, and any normalization technique will run into issues when we want a backward pass (e.g. sampling), i.e. there is no mean and variance for unnormalization. Currently, BN layer in flow models tends to register two buffers, i.e. running mean and running variance, to keep track of the batch mean and batch variance in the training, then fix it to generate data in the backward pass. We implement a GN layer for flow models similarly, i.e. registering two buffers to keep track of the group mean and variance in the training (we use 4 groups for each instance). However, during generation, we first try repeating the group mean and variance for the sampled batch, but the generation exhibits no variance, i.e. they all look similar. Then we change the registered buffers to keep track of a pool of group mean and variance, but the generation becomes null. We leave a tailored normalization layer that is suitable for private training for future work.

**Limitation:** The insight behind DP-LFlow, i.e. shrinking the model size, seems to contradict the intuition and the findings in a prior study on scaling laws of language models (Kaplan et al., 2020), because more

model parameters generally mean better learning capability. However, Kaplan et al. (2020) conducted all experiments in the non-private setting, and our findings in Figure 1 on image generators are consistent with Kaplan et al. (2020) in the non-private setting. In the private setting, Figure 1 motivates us to improve the performance of DPGMs by preventing the model from being over-complicated. Nevertheless, it is apparent that an over-simple generative model (certainly including DP-LFlow) is not able to generate realistic images either with or without DP guarantee. Therefore, shrinking the model size may impose an upper bound on the model utility.

## 6   Related Work

We categorize related works by approaches:

**DP-SGD:**   The vast majority of related works are realized by training different generative models with DP-SGD. **GAN:** DP-GAN (Xie et al., 2018) first trains GAN with DP-SGD algorithm, where the discriminator is trained with DP-SGD, then the generator is automatically DP as ensured by post-processing theorem. DP-CGAN (Torkzadehmahani et al., 2019) extends the idea into the conditional setting. **VAE:** DP-VaeGM (Chen et al., 2018) trains $k$ VAEs on $k$ classes of private data with the DP-SGD algorithm, and returns the union as generations. This work only focuses on privacy attacks. DP-kVAE (Acs et al., 2018) first partitions the dataset into $k$ clusters by kernel $k$-means method, then trains $k$ VAEs on each data cluster with DP-SGD. However, their generation exhibits clear mode-collapse. **Flow:** DP-NF (Waites & Cummings, 2021) directly trains a flow-based model by DP-SGD. DP-HFlow (Lee et al., 2022) designs a fine-grained gradient clipping strategy to increase the signal-to-noise ratio and accelerate the training. However, both works relating to flow models are limited to (low dimensional) tabular datasets. **Diffusion model:** A few recent papers also tried training the powerful diffusion models with DP-SGD (Dockhorn et al., 2023; Ghalebikesabi et al., 2023), which can generate realistic images under DP constraints. However, their training is significantly more computation-intensive, e.g. Dockhorn et al. (2023) requires 8 GPUs and one day to train a DP diffusion model on MNIST and FMNIST.

**PATE Mechanism:**   Private Aggregation of Teacher Ensembles (PATE) (Papernot et al., 2017; 2018) is another mechanism for learning a DP model, by perturbing the aggregated information from an ensemble of teacher models with noise. PATE-GAN (Jordon et al., 2019) first applies PATE mechanism to GAN, where the discriminator becomes non-differentiable, thus a student discriminator is trained with all teacher ensembles, which is then used to train the generator. G-PATE (Long et al., 2021) sanitizes the aggregated gradients from teacher discriminators to the generator to make the generator DP. However, gradient vectors need to be discretized in each dimension to employ the PATE mechanism that only takes categorical data as input. DataLens (Wang et al., 2021) further improves G-PATE by introducing a three-step gradient compression and aggregation algorithm called TopAgg.

**Kernel-based Methods:**   DP-MERF (Harder et al., 2021) proposes to perturb the kernel mean embeddings of real data through random Fourier features, and train a generator by minimizing the maximum mean discrepancy (MMD) between the noisy embedding of private input and embedding of generation. PEARL (Liew et al., 2022) extends the idea of DP-MERF by introducing an adversarial objective on sampling frequencies, which indicates improvement in performance.

**Others:**   GS-WGAN (Chen et al., 2020) creates a privacy barrier at the output of the generator of a Wasserstein GAN (WGAN), based on the observation that only the generator will be published, thus only the generator needs to be private. DP-Sinkhorn (Cao et al., 2021) adds a privacy barrier in a similar way as GS-WGAN, where a Sinkhorn loss is used as the training objective.

## 7   Conclusion

Though DP-SGD is currently the workhorse algorithm for training a deep learning model, it remains a big challenge whether it can be applied to training large models with acceptable model utility. In this paper,

we first show that training a DP flow via DP-SGD is highly challenging (or even infeasible) with achieving acceptable utility due to a few particular challenges of flow models, and then propose an effective solution, i.e. DP-LFlow, by reducing the flow training from the full input space to a lower dimensional latent space, so that the model is more resilient to (larger) noise perturbation introduced by DP-SGD. Experimental results on widely compared image benchmarks demonstrate the generality and scalability of DP-LFlow on different image spaces (grayscale and RGB) and different DP constraints (weak and strong DP guarantees). Notably, to our best knowledge, DP-LFlow is the first DPGM to scale to high-resolution image datasets, which further validates its effectiveness and versatility.

**Acknowledgments**

We thank all the reviewers and the action editor for thoughtful comments. We gratefully acknowledge funding support from NSERC and University of Waterloo, and the computing resources from the Vector Institute.

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

## A Proof

**Theorem 3.1.** Let $\mathcal{M}_i$ $(i = 1, 2, \ldots, k)$ be $k$ DP mechanisms, and each $\mathcal{M}_i$ satisfies $(\epsilon_i, \delta_i)$-DP. Given a deterministic partitioning function $f$, let $D_1, D_2, \ldots, D_k$ be the disjoint partitions by executing $f$ on $D$. Releasing $\mathcal{M}_1(D_1), \ldots, \mathcal{M}_k(D_k)$ satisfies $(\max_{i \in \{1, 2, \ldots, k\}} \epsilon_i, \max_{i \in \{1, 2, \ldots, k\}} \delta_i)$-DP.

*Proof.* Without loss of generality, given two neighboring datasets $D$ and $D'$, assume that $D$ contains one more element than $D'$. Executing $f$ on $D$ and $D'$, we have partitions $D_1, D_2, \ldots, D_k$ and $D'_1, D'_2, \ldots, D'_k$, respectively. There exists $j$ such that (1) $D_j$ contains one more element than $D'_j$, and (2) $D_s = D'_s$ for $s = 1, 2, \ldots, k$ and $s \neq j$. Denote $\mathcal{M}_1(D_1), \ldots, \mathcal{M}_k(D_k)$ by $\mathcal{M}(D)$. Since the subsets are disjoint from each other, running $k$ algorithms on each subset is independent of each other. For any sequence $t = (t_1, t_2, \ldots, t_k)$ of outputs of $\mathcal{M}_1, \ldots, \mathcal{M}_k$ where $t_i \in Range(\mathcal{M}_i)$, we have:

$$\Pr[\mathcal{M}(D) = t] = \Pr[\mathcal{M}_1(D_1) = t_1 \wedge \mathcal{M}_2(D_2) = t_2 \wedge \ldots \wedge \mathcal{M}_k(D_k) = t_k] \tag{6}$$

$$= \Pr[\mathcal{M}_j(D_j) = t_j] \prod_{s=1,2,\ldots,k,s \neq j} \Pr[\mathcal{M}_s(D_s) = t_s] \tag{7}$$

$$\leq \left( \exp(\epsilon_j)\Pr[\mathcal{M}_j(D'_j) = t_j] + \delta_j \right) \prod_{s=1,2,\ldots,k,s \neq j} \Pr[\mathcal{M}_s(D'_s) = t_s] \tag{8}$$

$$= \exp(\epsilon_j) \prod_{i=1,2,\ldots,k} \Pr[\mathcal{M}_i(D'_i) = t_i] + \delta_j \prod_{s=1,2,\ldots,k,s \neq j} \Pr[\mathcal{M}_s(D'_s) = t_s] \tag{9}$$

$$= \exp(\epsilon_j)\Pr[\mathcal{M}(D') = t] + \delta_j \prod_{s=1,2,\ldots,k,s \neq j} \Pr[\mathcal{M}_s(D'_s) = t_s] \tag{10}$$

$$\leq \exp(\epsilon_j)\Pr[\mathcal{M}(D') = t] + \delta_j \tag{11}$$

$$\leq \exp(\max_{i=1,2,\ldots,k} \epsilon_i)\Pr[\mathcal{M}(D') = t] + \max_{i=1,2,\ldots,k} \delta_i \tag{12}$$

$\square$

## B Datasets

We briefly introduce the public datasets and associated preprocessing. Image size is shown in $\#channels \times height \times width$. Images are normalized to the range of $[0, 1]$.

**MNIST (LeCun et al., 1998) & Fashion MNIST (Xiao et al., 2017):** MNIST contains hand-written digits images, whereas Fashion MNIST contains cloth and shoe images. Images in both datasets are single-channel, in the size of $1 \times 28 \times 28$, and have 10 classes. We adopt the official training and test split. 10k images from the training split are randomly held out as the validation set.

**CelebA (Liu et al., 2015):** CelebA is a dataset including face images of celebrities. Each image is in the size of $3 \times 178 \times 218$ and has 40 binary attributes. All images are cropped to $3 \times 178 \times 178$, and then resized to $3 \times 32 \times 32$. We also adopt the official training, validation and test split, but randomly select 50k images of each gender from the training split as our training set.

**CelebA-HQ (Karras et al., 2018):** CelebA-HQ is a high-quality version of CelebA, which is commonly recognized as a high-resolution image set. It consists of 30k images in total. We download a gender conditioned split (where images are resized to $3 \times 256 \times 256$) by following this link. 1999 images are randomly held out from the training split as the validation set.

## C Rényi differential privacy (RDP)

Rényi differential privacy (RDP) extends ordinary DP using Rényi's $\alpha$ divergence (Rényi, 1961) and provides tighter and easier composition property than the ordinary DP notion, thus we adopt RDP to accumulate the privacy cost. Formally, we recall:

Table 5: Network configurations for different datasets in the experiments. #*h_conv* denotes the number of hidden sizes in the convolutional layers. #*h_lin* denotes the number of hidden sizes in the linear layers. #*c* denotes the length of latent code. #*b* means the number of blocks in flow.

| Dataset | #*h_conv* in encoder | #*h_conv* of decoder | #*c* | #*b* | #*h_lin* of flow |
|---|---|---|---|---|---|
| MNIST | [32, 64] | [64, 32] | 20 | 9 | 200 |
| FMNIST | [32, 64] | [64, 32] | 20 | 9 | 200 |
| CelebA | [64, 128, 256] | [256, 128, 64] | 32 | 9 | 200 |
| CelebA-HQ | [16, 32, 64, 128, 256, 512] | [512, 256, 128, 64, 32, 16] | 64 | 12 | 256 |

**Definition C.1** (($\alpha, \epsilon$)-RDP (Mironov, 2017)). A randomised mechanism $\mathcal{M}$ is ($\alpha, \epsilon$)-RDP if for all adjacent inputs $D, D'$, Rényi's $\alpha$-divergence (of order $\alpha > 1$) between the distribution of $\mathcal{M}(D)$ and $\mathcal{M}(D')$ satisfies:

$$\mathbb{D}_\alpha(\mathcal{M}(D)\|\mathcal{M}(D')) := \tfrac{1}{\alpha-1} \log \mathbb{E}_{Z\sim Q} \left( \tfrac{P(Z)}{Q(Z)} \right)^\alpha \leq \epsilon, \tag{13}$$

where $P$ and $Q$ are the density of $\mathcal{M}(D)$ and $\mathcal{M}(D')$, respectively (w.r.t. some dominating measure $\mu$).

Importantly, a mechanism satisfying ($\alpha, \epsilon$)-RDP also satisfies($\epsilon + \frac{\log 1/\delta}{\alpha-1}, \delta$)-DP for any $\delta \in (0,1)$.

Conveniently, RDP is linearly composable:

**Theorem C.1** (Sequential composition of RDP (Mironov, 2017)). If mechanism $\mathcal{M}_i$ satisfies ($\alpha, \epsilon_i$)-RDP for $i = 1, 2, \ldots, k$, then releasing the composed mechanism $(\mathcal{M}_1, \ldots, \mathcal{M}_k)$ satisfies ($\alpha, \sum_{i=1}^k \epsilon_i$)-RDP.

We also adopt the Gaussian mechanism for achieving RDP:

**Definition C.2** (Gaussian mechanism for RDP (Dwork et al., 2014; Mironov, 2017)). Let $f : \mathcal{D} \to \mathbb{R}^p$ be an arbitrary $p$-dimensional function with sensitivity:

$$\Delta_2 f = \max_{D,D'} \|f(D) - f(D')\|_2 \tag{14}$$

for all adjacent datasets $D, D' \in \mathcal{D}$. The Gaussian mechanism $\mathcal{M}_\sigma$ perturb the output of $f$ with Gaussian noise:

$$\mathcal{M}_\sigma = f(D) + \mathcal{N}(0, \sigma^2 \cdot \mathbb{I}) \tag{15}$$

where $\mathbb{I}$ is identity matrix. Then, $\mathcal{M}_\sigma$ satisfies ($\alpha, \frac{\alpha(\Delta_2 f)^2}{2\sigma^2}$)-RDP.

DP-SGD tracks the total privacy consumption as follows: (1) for each training iteration, compute the RDP privacy cost for a subsampled batch where Gaussian mechanism is applied; (2) compose RDP mechanisms over training iterations; (3) convert RDP back to ($\epsilon, \delta$)-DP. The implementation details are given below.

## D   Framework

The schematic of DP-LFlow is shown in Figure 7.

## E   Implementation

### E.1   Flow models

Our code for flow generative models are adapted from public repos, i.e. Glow and RealNVP. Hyperparameters of the network are selected by comparing the performance on the validation set, and the selection results are given in Table 5.

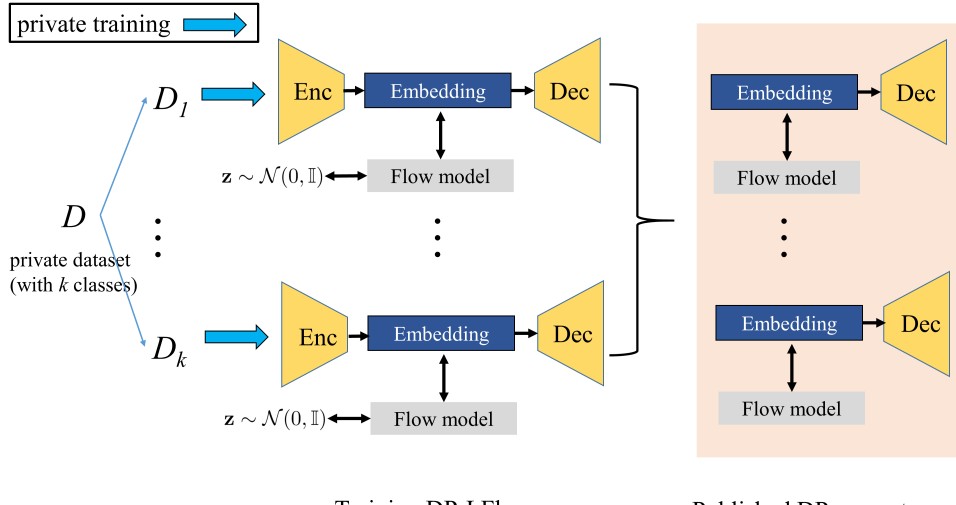

Figure 7: The framework of DP-LFlow.

## E.2   Privacy implementation

We use a public repo, i.e. pyvacy, for implementing DP-SGD algorithm, as well as the total privacy calculation. Pyvacy tracks the privacy loss by RDP accountant, which is a PyTorch implementation based on Tensorflow Privacy.

For all datasets we use, we set subsampling rate as 0.1, training iterations as 300, noise multiplier as 1.25 to target $(10, 10^{-5})$-DP and 4.5 to target $(1, 10^{-5})$-DP, respectively. With better evaluation performance on the validation set, gradient clipping norms are set as 0.1 for MNIST and Fashion MNIST, 0.01 for CelebA, and 10 for CelebA-HQ.

## E.3   Fréchet Inception Distance (FID)

FID calculates the distance between the feature vectors extracted by InceptionV3 pool3 layer (Szegedy et al., 2016) on real and synthetic samples. Specifically,

$$\text{FID} = \|\mu_r - \mu_g\|_2^2 + \text{Tr}(\Sigma_r + \Sigma_g - 2(\Sigma_r \Sigma_g)^{\frac{1}{2}}) \tag{16}$$

where $X_r \sim \mathcal{N}(\mu_r, \Sigma_r)$ and $X_g \sim \mathcal{N}(\mu_g, \Sigma_g)$ are activations of InceptionV3 pool3 layer of real images and generated images, respectively, and $\text{Tr}(A)$ refers to the trace of a matrix $A$. Intuitively, a lower FID means the generation $X_g$ is more realistic (or more similar to $X_r$). We use a PyTorch implementation for computing FID, which will resize images and repeat channels three times for grayscale images to meet the input size requirement.

## E.4   Classification task

We follow Cao et al. (2021) for the classifier implementation. We import scikit-learn package for implementation logistic regression classifier (e.g. from sklearn.linear_model import LogisticRegression) with default parameter settings.

The MLP network consists of following layers: linear($input\_dim, 100$) $\rightarrow$ ReLU $\rightarrow$ linear($100, output\_dim$) $\rightarrow$ Softmax.

The CNN consists of following layers: Conv2d($input\_channels$, 32, kernel_size=3, stride = 2, padding=1) $\rightarrow$ Dropout(p=0.5) $\rightarrow$ ReLU $\rightarrow$ Conv2d(32, 64, kernel_size=3, stride = 2, padding=1) $\rightarrow$ Dropout(p=0.5) $\rightarrow$ ReLU $\rightarrow$ flatten $\rightarrow$ linear($flatten\_dim, output\_dim$) $\rightarrow$ Softmax.

Both MLP and CNN are optimized by Adam with default parameters. All classifiers are trained on synthetic data, and we report test accuracy on real test data as the evaluation metric.

### E.5 Baselines

All results of DP-MERF (Harder et al., 2021) are obtained by running their code with default parameters. It is worth mentioning that DP-MERF does not implement on CelebA. We adapt their code on CelebA by using the generative network they designed for SVHN with $16, 8, 8$ channels for three convolutional layers, respectively.

GS-WGAN only implements on $(10, 10^{-5})$-DP. To target $(1, 10^{-5})$-DP, we tried two vanilla variations by tuning parameters of $(10, 10^{-5})$-DP in their code, i.e. increasing noise scale while keeping the rest parameters unchanged, or decreasing the number of iterations while keeping the rest parameters unchanged. We experimentally found that both variations will not generate meaningful images. The former variation is not even able to generate anything on Fashion MNIST, so we instead present the latter variation for comparison.

All other results (e.g. numbers in the tables, generated images) are cited from papers as we specify.

