# OpenReview forum: "DP-LFlow: Differentially Private Latent Flow for Scalable Sensitive Image Generation"
_TMLR — Accepted by TMLR_

### Review · Reviewer_8NY2 · 2023-07-06

**Summary Of Contributions:**

The paper proposes a new method for differentially private data generation, using normalizing flow. A few tricks are used to make DP training on normalizing flow work, one is to use latent flow, another is to replaced the conditional generation with different models for different classes.

**Audience:**

Yes

**Broader Impact Concerns:**

No concern.

**Claims And Evidence:**

Yes

**Requested Changes:**

The key change in the paper seems to be using a separate model for each class, this would be impractical when number of classes is large, have the authors considered other alternatives? Adding an in depth discussion on this design choice would be highly appreciated.

**Strengths And Weaknesses:**

Strengths:
1. The performance is state-of-the-art among the compared baselines.
2. The proposed method is easy to understand and implement

Weaknesses:
1. The paper does seem lack novelty since it is just applying DP to latent flow models, and having one model for each class is intuitively straightforward and impractical when number of classes is large, or even infinite (e.g., text-guided image generation).

---

> ### Author Response · Authors · 2023-07-14
> **Response to reviewer 8NY2**
>
> > Q1. The paper does seem lack novelty since it is just applying DP to latent flow models.
>
> **R1:** For the novelty concern, we would like to explain from the following perspectives:
> + **DP flow on image sets is novel:** As we mentioned in the introduction, while there are plenty of GAN-based models in developing DPGMs, the method relating to normalizing flow is comparatively limited and restricted to lower-dimensional tabular dataset. The capability of exact density estimation makes flow distinct from other generative models wherever applications want to explore density. Our Section 4.4 gives such as example.
> + **Training DP flow is not easy:** We want to note that directly applying DP-SGD to flow model is not going to work, even on small image datasets (see our Section 4.5.1 for details). Any ML researcher/engineer will encounter the two non-trivial empirical challenges. Our work intends to seek an efficient approach to obtain a better privacy-utility trade-off by training a DP-flow on high-dimensional dataset, which indeed achieves this goal.
> + **DP-LFlow is scalable to a variety of datasets:** Moreover, unlike a few SOTA works (e.g. GS-WGAN) that only run experiments on simpler datasets (i.e. MNIST and Fashion MNIST), we evaluate our proposed method on more complex datasets (e.g. CelebA and CelebA-HQ), to further validate the effectiveness and scalability of DP-LFlow. Notably, our method, to the best of our knowledge, is the first DPGM that present meaningful generation with privacy constraints on real high-resolution images sets (where input size is up to 3x256x256). The empirical novelty of scaling DP generation from 1x28x28 to 3x256x256 is (arguably) non-trivial.
>
> We expect this work to enrich the family of DPGMs and serve as a reference for ML practitioners who are interested in training a DP-flow, for its effectiveness and scalability across a variety of datasets and privacy guarantees.
>
> > Q2. Having one model for each class is intuitively straightforward and impractical when number of classes is large, or even infinite (e.g., text-guided image generation). Have the authors considered other alternatives? Adding an in-depth discussion on this design choice would be highly appreciated.
>
> **R2:** Thanks for your question! We agree that the benefits of partitioning the dataset will come at some costs, such as what you mentioned. Nevertheless, we believe two major benefits outweigh the drawback:
> + **Shrinking model size.** As we repeatedly mentioned, smaller models are more resilient to noise perturbations. Partitioning the dataset allows each generator to be only interactive with a sole data modality instead of multi-modalities, and thereby requires less model complexity.
> + **Training efficiency.** The significant training time overhead remains a notable challenge for DP-SGD due to the gradient clipping and randomization. Partitioning the dataset allows us to train each sub-model in parallel, which is more efficient.
> + As suggested by prior work [1], a larger batch size contributes to better utility for a DP image classifier. Partitioning the dataset allows us train each sub-model with larger batch sizes with limited GPU memory.
>
> We will include a discussion on limitations in our draft. As for our design choice, we do compare it with the one-model-for-all paradigm, which is given in our second ablation study, i.e. section 4.5.2. Table 4 clearly shows that the one-model-per-class paradigm (With partitioning the dataset) achieves better model utility compared to the one-model-for-all paradigm (Without partitioning the dataset). We will make this point more clear in our manuscript.
>
> [1] De, S., Berrada, L., Hayes, J., Smith, S. L., & Balle, B. (2022). Unlocking high-accuracy differentially private image classification through scale. arXiv preprint arXiv:2204.13650.

---

### Review · Reviewer_2bxJ · 2023-07-06

**Summary Of Contributions:**

The paper proposes a strategy called DP latent flows to do image generation, which trains a normalizing flow on a latent space, to generate images. This also allows the generative model to produce a density estimate, which can be used for tasks such as anomaly detection. They evaluate the performance of the approach on small image datasets, as well as CelebA.

**Audience:**

Yes

**Claims And Evidence:**

Yes

**Requested Changes:**

Discuss work on DP diffusion models at least in the related work.

**Strengths And Weaknesses:**

Strengths:
The paper scales DP generative models to image datasets with reasonable DP parameters. Downstream training is quite accurate using this approach.

The approach permits density estimation, allowing for OOD detection.

The paper is clearly written.

Weaknesses:

The contributions are limited by related work on DP diffusion models and Lee et al. which is cited. The work on DP diffusion models is also capable of generating reasonable images which lead to strong downstream performance [r1 r2].

The way the paper performs conditional generation is by using a new model for each partition of the dataset. This is a reasonable fix for the small datasets considered, but will run into challenges when training models on more diverse data, such as classifier or encoder-guided generation as is done in state of the art nonprivate image generation.

Overall review:

I recommend accepting the paper. The main issue with it is a lack of comparison to DP diffusion models.

Claims and evidence: I think the claims made by the paper are mostly accurate, but the paper omits a comparison to recent work on DP diffusion models. I would like to see a comparison to this work.

Audience: Many people in the community are interested in training differentially private generative models. I think this paper will be interesting to them.

[r1] https://arxiv.org/abs/2210.09929

[r2] https://arxiv.org/abs/2302.13861

---

> ### Author Response · Authors · 2023-07-14
> **Response to Reviewer 2bxJ**
>
> > Q1. The contributions are limited by related work on DP diffusion models and Lee et al. which is cited. The work on DP diffusion models is also capable of generating reasonable images which lead to strong downstream performance [r1 r2]. I would like to see a comparison to this work; Discuss work on DP diffusion models at least in the related work.
>
> **R1:** Thank you for bringing the important references into our attention! We check the references and found that [r1,r2] indeed can generate realistic images with SOTA numerical evaluations. Specifically, [r1] yields classification accuracy as high as 98.1 on MNIST when $\epsilon=10$, whereas [r2] improves [r1] by using pre-trained models and a few other practical training tricks (which require intensive computing resources) so they achieve 98.6. [r2] did not report results on Fashion MNIST and CelebA, so our following comparison will go with [r1]. The classification accuracies of [r1] on MNIST and FMNIST when $\epsilon=1$ are 93.4 and 73.6, respectively, while ours are 88.9 and 77.4, respectively. On CelebA, they only did unconditional generation, while we still do conditional generation. More importantly, DP-LFlow still has **two unique properties** compared to [r1,r2]: (1) DP-LFlow is able to conduct the DP-OOD detection task; (2) DP-LFlow can generate large images up to 3x256x256, while both [r1] and [r2] only present generation results on image sets up to 3x32x32.
>
> Besides model utility, our method is **way less computation-intensive** than both [r1] and [r2]. [r2] requires more computational resources than [r1], so the following comparison will still go with [r1]. [r1] requires **8 GPUs and one day** to train a DP diffusion model on MNIST and FMNIST, while our method only requires **1 single GPU and a few hours**; On CelebA, [r1] needs **8 GPUs and 4 days**, while our method only needs **1 GPU and around half-day**.
>
> We will add the above discussion to our draft.
>
> > Q2. The way the paper performs conditional generation is by using a new model for each partition of the dataset. This is a reasonable fix for the small datasets considered, but will run into challenges when training models on more diverse data, such as classifier or encoder-guided generation as is done in state of the art nonprivate image generation.
>
> **R2:** Thanks for your question! We agree that the benefits of partitioning the dataset will come at some costs, such as what you mentioned. Nevertheless, we believe two major benefits outweigh the drawback:
> + **Shrinking model size.** As we repeatedly mentioned, smaller models are more resilient to noise perturbations. Partitioning the dataset allows each generator to be only interactive with a sole data modality instead of multi-modalities, and thereby requires less model complexity.
> + **Training efficiency.** The significant training time overhead remains a notable challenge for DP-SGD due to the gradient clipping and randomization. Partitioning the dataset allows us to train each sub-model in parallel, which is more efficient.
> + As suggested by prior work [1], a larger batch size contributes to better utility for a DP image classifier. Partitioning the dataset allows us train each sub-model with larger batch sizes with limited GPU memory.
>
> We will include a discussion on limitations in our draft.
>
> [1] De, S., Berrada, L., Hayes, J., Smith, S. L., & Balle, B. (2022). Unlocking high-accuracy differentially private image classification through scale. arXiv preprint arXiv:2204.13650.

---

### Review · Reviewer_CidX · 2023-07-24

**Summary Of Contributions:**

This paper explores the process of training flow-based generative models in the context of differential privacy. It is inspired by the recent success of diffusion model. Initially, this work applies an encoder to reduce the dimensionality of the input image and transforms the input image into some latent space. The generative model is then learned to generate latent codes that are distributed similar to those mapped from real data. According to the experimental results, the proposed method significantly outperforms the existing state-of-the-art DP generative models.

**Audience:**

Yes

**Claims And Evidence:**

Yes

**Requested Changes:**

1. On page 2: "flow model usually apply BN model ... where the per-example gradient is not available". Could you please provide more clarify on why per-example gradient is not available in this case. My understanding is that BN shares statistical information across examples, but it does not preclude the computation of the per-example gradient.
2. The baseline flow models are proposed more than five years ago. Could you explain the choice of these specific models, why not choosing some more recent models?

**Strengths And Weaknesses:**

Strengths:
1. The authors have written this work very well, clearly elucidating their motivation, techniques and implementation details.
2. From an empirical standpoint, this work outperforms previous works, indicating considerable progress.
3. The concept, although simple, proves effective. This work is notably the first to achieve DP generation on high-resolution data.

Weakness:
My major concern is that the method proposed in this work looks like a provisional solution to the challenge of DP to me. It does not genuinely address the problem in the practical application of DP, specifically the performance degradation of large network due to the added noise. It is clear now that large network can bring more benefits, we can also see it from the Figure 1 in this paper. While reducing the network size can enhance DP training efficiency, constrained network size also caps the performance that can be potentially obtained, which may prevent such frameworks from achieving satisfying results in the long run. Give this limitation, I question a little the future prospects of studying DP in this particular manner.

---

> ### Author Response · Authors · 2023-07-25
> **Response to Reviewer CidX**
>
> > Q1. My major concern is that the method proposed in this work looks like a provisional solution to the challenge of DP to me. It does not genuinely address the problem in the practical application of DP, specifically the performance degradation of large network due to the added noise. It is clear now that large network can bring more benefits, we can also see it from the Figure 1 in this paper. While reducing the network size can enhance DP training efficiency, constrained network size also caps the performance that can be potentially obtained, which may prevent such frameworks from achieving satisfying results in the long run. Give this limitation, I question a little the future prospects of studying DP in this particular manner.
>
> **Response:** Thanks for your comment. Indeed, we mentioned that smaller models are more resilient to noise perturbation, but we also mentioned that too simple model is not desired either (in the beginning of page 5). We need the model to be small but expressive enough to properly learn the input distribution. Inspired by the fact that the semantic meaning of images still remains after aggressive compression (see Figure 2 in [Rombach et al., 2022]), we propose to train a normalizing flow in the latent space. Figure 1 indeed shows that in the non-private setting, a larger VAE (75.5 MB) can achieve better FID ($\approx$ 25 at best, further increasing the model complexity does not improve much). As we also mentioned on page 5, the proposed latent flow (only 2.5 MB, almost 30 times smaller) in the non-private setting achieves an FID as low as 12.5, which outperforms the larger VAE by a large margin and is an instance of a small yet expressive enough generative model. Therefore, instead of directly improving the DP training of raw large normalizing flows, we show that  a latent flow benefits from both training efficiency and the model utility, which indicates its promising use in the DP setting.
>
> > Q2. On page 2: "flow model usually apply BN model ... where the per-example gradient is not available". Could you please provide more clarify on why per-example gradient is not available in this case. My understanding is that BN shares statistical information across examples, but it does not preclude the computation of the per-example gradient.
>
> **Response:** Thanks for your question. The BN layer in normalizing flow is supposed to compute BN statistics (e.g. mean $\mu$ and variance $var$ ) for each batch in the training, and use the statistics to center and rescale a batch. However, when computing the per-example gradient in DP-SGD, the effective 'batch size' becomes 1 for each example in a batch. Since the variance of a single input is 0, the batch normalization, e.g.  $x = \frac{x-\mu}{\sqrt{var}}$ will be $\infty$ for every example, which also destroys the gradient computation. People generally add a very small number to $var$ to avoid numerical stability issues, but it will not change the fact that the per-example gradient is not meaningful.

---

> > ### Author Response · Authors · 2023-07-25
> > **Continued...**
> >
> > > Q3. The baseline flow models are proposed more than five years ago. Could you explain the choice of these specific models, why not choosing some more recent models?
> >
> > **Response:** Thanks for your question. We primarily want to pick normalizing flows that are representative and suitable for DP training, so that we don't need to try all existing normalizing flows (not possible either) to validate our DP latent flow idea. Both RealNVP and Glow are still representative normalizing flows, and they are widely compared and used in existing related works and in practice. They are still able to produce SOTA (or comparable to SOTA) performance in certain tasks.
> >
> > We do note that there are some recent papers on normalizing flows [1,2,3,4], but not every paper focuses on image generation as us. For example, [2] is for 3D objects, [3] is for time series data, and [4] is for graphs. [1] seems to be the most related one, which falls into the category of residual flows (based on ResNet). Compared to coupling layer-based flows (e.g. RealNVP and Glow), residual flows are more flexible by lifting the architecture restriction and are shown to be (slightly) better in some tasks. For example, according to [1], RealNVP and Glow yield 1.06 and 1.05 bpd (bits-per-dimension, lower is better) on MNIST, 3.49 and 3.35 on CIFAR-10, whereas [1] produces 0.93 bpd on MNIST and 3.22 on CIFAR10.
> >
> > However, residual flows are numerically much harder to compute and slow in the training, and this training time overhead can be amplified by DP-SGD. To give you the context, [1] claims to use 4 RTX 3090 GPUs on MNIST and up to 8 A100 GPUs on RGB datasets (32x32 or 64x64 images). As they don't mention training time, it remains unclear whether one can train a residual flow with limited GPUs (e.g. a single GPU) and how long to train it. On the contrary, coupling layers are cheap to compute, e.g. one can easily train an RealNVP with one single GPU in reasonable time.
> >
> > Therefore, by taking representativeness, performance, computation restrictions, and training time overhead by DP-SGD all into consideration, we now adopt RealNVP and Glow in this work, but we do not rule out the possibility of improving DP-LFlow if there is a better flow model in the future that fits our setting.
> >
> > [1] Ahn, B., Kim, C., Hong, Y., \& Kim, H. J. (2022). Invertible Monotone Operators for Normalizing Flows. Advances in Neural Information Processing Systems, 35, 16836-16848.
> >
> > [2] Postels, J., Danelljan, M., Van Gool, L., \& Tombari, F. (2022). ManiFlow: Implicitly Representing Manifolds with Normalizing Flows. In 2022 International Conference on 3D Vision (3DV) (pp. 84-93). IEEE.
> >
> > [3] Feng, S., Xu, K., Wu, J., Wu, P., Lin, F., \& Zhao, P. (2022). Multi-scale attention flow for probabilistic time series forecasting. arXiv preprint arXiv:2205.07493.
> >
> > [4] Wehenkel, A., \& Louppe, G. (2021). Graphical normalizing flows. In International Conference on Artificial Intelligence and Statistics (pp. 37-45). PMLR.

---

> > > ### Comment · Reviewer_CidX · 2023-08-13
> > > **Final response**
> > >
> > > Regarding Q1, I'm not fully convinced by the authors' response. My reservation could be alleviated if the authors can provide a plot illustrating the scaling laws ([1] is a reference for such study in non-private settings) under this framework, across a wide range of network sizes and dataset sizes. Otherwise, for the sake of completeness, I believe it is essential to explicitly state in the limitation section, that the proposed framework may impose an upper bound on performance since this framework specifically discourages the network from scaling up.
> > >
> > > If this concern is addressed, I recommend accepting this work.
> > >
> > > [1] Scaling Laws for Neural Language Models. Jared Kaplan et al., 2020

---

> > > > ### Author Response · Authors · 2023-08-13
> > > >
> > > > Thanks for the great question! We believe the current Figure 1 is such a plot illustrating the scaling law (on network size) for VAE, where a larger VAE performs better in the non-private setting but worse in the private setting, which motivates us to reduce the size for a generative model (GM). However, it is apparent that if the network is too simple, the GM (either with or without DP guarantee) is not able to generate good images, and this intuition is true for our DP-LFlow as well, so we agree that larger networks for a GM are more capable of learning the input distribution. One of the messages we want to convey is that the GM with DP guarantee is not necessary to be too complex to achieve a better privacy-utility tradeoff.
> > > >
> > > > We also agree that we should put this point in an explicit limitation section, which we will do in the revision.

---

### Decision · Action_Editors · 2023-09-18

**Recommendation:** Accept with minor revision

**Comment:**

The reviewers were generally positive, and felt this was an important problem.
A main issue raised was about the lack of comparisons with recent works on DP diffusion models. The authors discussed this in their rebuttal, and promised to add some additional results and discussions contrasting their work with those on DP diffusion models.
Another point to add in the revision is the discussion with reviewer CidX about the claimed scaling laws.
These should be added for the final version.

**Audience:**

Yes, there is a large audience interested in privacy, generative models, and their intersection.

**Claims And Evidence:**

The claims are overall accurate and convincing.
Some reviewers commented that comparisons to alternative approaches such as DP diffusion models could be warranted and I agree.
The authors promised to include some additional discussions and results along these lines in the rebuttal, and these should be included for a final version.